# Microstructure and Its Effect on the Magnetic, Magnetocaloric and Magnetostrictive Properties of Tb_55_Co_30_Fe_15_ Glassy Ribbons

**DOI:** 10.3390/ma14113068

**Published:** 2021-06-04

**Authors:** Xin Wang, Kang-Cheung Chan, Lei Zhao, Ding Ding, Lei Xia

**Affiliations:** 1College of Engineering, Shanghai Polytechnic University, Shanghai 201209, China; wangxin2020@126.com; 2Advanced Manufacturing Technology Research Centre, Department of Industrial and Systems Engineering, The Hong Kong Polytechnic University, Hung Hom, Hong Kong; 13901872r@connect.polyu.hk; 3Institute of Materials, Shanghai University, Shanghai 200072, China; d.ding@shu.edu.cn

**Keywords:** metallic glass, microstructure, magnetocaloric effect, magnetostriction

## Abstract

In the present work, the microstructure and its effect on the magnetic, magnetocaloric, and magnetoelastic properties of the Tb_55_Co_30_Fe_15_ melt-spun ribbon were investigated. The ribbon exhibits typical amorphous characteristics in its X-ray diffraction examination and differential scanning calorimetry measurement. However, the magnetic properties of the ribbon indicate that the ribbon is inhomogeneous in the nanoscale, as ascertained by a high-resolution electron microscope. Compared to the Tb_55_Co_45_ amorphous alloy, the Tb_55_Co_30_Fe_15_ ribbon shows poor magnetocaloric properties but outstanding magnetostriction. A rather high value of reversible magnetostriction up to 788 ppm under 5 T was obtained. The mechanism for the formation of nanoparticles and its effect on the magnetocaloric and magnetostrictive properties were investigated.

## 1. Introduction

Magnetostriction, which refers to the change in shape or dimension in response to a magnetic field, is one of the inherent properties of magnetic materials that was firstly discovered in 1842 by J. P. Joule in Fe [1,2]. These magnetic materials can be applied as sensors, actuators, and energy harvesting devices because the magnetostrictive response allows these ferromagnetic materials to convert electromagnetic energy to mechanical energy. Therefore, magnetostrictive materials have evoked increasing interest in recent years, especially when the giant magnetostriction effect was found in TbDyFe alloy (known as Terfenol-D) [3,4]. The TbDyFe alloy exhibits ultrahigh magnetostriction up to 2000 ppm. However, the shortcomings inherited in the Terfenol-D alloy, such as brittleness, poor corrosion properties, and high eddy-current loss, can hardly be overcome in practical applications. The Cu addition can improve the fracture toughness of the TbDyFe alloy, but unfortunately deteriorate the magnetostriction performance of the alloy [5].

The vitrification of the TbDyFe alloy may provide a useful way to overcome the above shortcomings: by enhancing the strength of the alloy and making the glassy alloy less brittle; by improving the corrosion resistance; and by decreasing the eddy-current loss through enlarging the electric resistance. Although the magnetostriction of the amorphous (Tb_0.3_Dy_0.7_)_40_Fe_60_ thin film (λ = 400 ppm) is not as high as the Terfenol-D alloy [6], it is still comparable or higher than the Gafenal alloys [7,8,9]. However, the (Tb_0.3_Dy_0.7_)_40_Fe_60_ amorphous alloy can only be fabricated into the shape of thin film due to its poor glass forming ability (GFA). Recently, we systematically investigated the GFA and magnetic properties of binary Tb-transition metal (TM) and Dy-TM alloys and obtained excellent magnetostrictive properties in these binary amorphous ribbons. For example, the Tb_62.5_Co_37.5_ fully amorphous alloy exhibits a rather high magnetostriction: ~320 ppm and ~470 ppm under 2 T and 5 T, respectively, at 50 K [10]; while the Dy_50_Co_50_ glassy ribbon shows a higher magnetostriction: ~320 ppm and ~600 ppm under 2 T and 5 T, respectively, at 60 K [11].

More recently, we replaced the Co element with Fe in the binary Tb-Co amorphous alloys for further improving its glass formability and enhancing its Curie temperature and magnetostriction. In the present work, a high magnetostriction of over 700 ppm in a Tb_55_Co_30_Fe_15_ glassy ribbon is reported. Although the ribbon shows amorphous characteristics in its X-ray diffraction (XRD) pattern and the differential scanning calorimetry (DSC) curve, the results of the magnetic measurements suggest that the as-spun ribbons may be inhomogeneous in nanoscaled microstructures. The effect of the microstructure on the magnetocaloric and magnetoelastic properties was investigated.

## 2. Materials and Methods

A Tb_55_Co_30_Fe_15_ ingot was produced by arc-melting pure Tb, Co, and Fe metals, purchased from the Trillion Metals Co., Ltd. (Beijing, China) with the purities over 99.9% (at%), by a non-consumable electrode in a high vacuum furnace. Ribbons were fabricated by injecting the melt of the Tb_55_Co_30_Fe_15_ alloy from a quartz tube to a rotating copper roller under the protection of pure Ar. The structure of the ribbons was examined by a Rigaku diffractometer (D\max-rC, Cu *K_α_* radiation). The glass transition and crystallization behaviors of the glassy sample were observed on a DSC curve of the Tb_55_Co_30_Fe_15_ ribbon measured at a heating rate of 0.667 K/s on a NETZSCH calorimeter (model 404C, Selb, Germany). Microstructural observation of the Tb_55_Co_30_Fe_15_ ribbon was performed on a JEOL field emission high resolution electron microscope (HREM, model JEM-2010F, Tokyo, Japan). The sample for HREM observation was prepared by electrolytic polishing under the protection of liquid nitrogen. Magnetic properties of the glassy sample were measured by a vibrating sample magnetometer (VSM) module in a Physical Properties Measurement System (PPMS, Ever cool II, Quantum Design, San Diego, CA, USA). Magnetostriction (*λ*) of the Tb_55_Co_30_Fe_15_ ribbon, as well as the Tb_55_Co_45_ ribbon for comparison purposes, was measured by PPMS using a foil strain gauge (KYOWA; model KFL-02-120-C1, Tokyo, Japan), which was calibrated by pure aluminum.

## 3. Results

Figure 1 shows the X-ray diffraction (XRD) pattern of the Tb_55_Co_30_Fe_15_ ribbon and the DSC curve of the ribbon. The broad diffraction hump in the XRD pattern (Figure 1a), the endothermic glass transition and the sharp exothermic crystallization in the DSC trace (Figure 1b) illustrate the typical amorphous characteristics of the Tb_55_Co_30_Fe_15_ ribbon. The glass transition temperature (*T_g_*), crystallization temperature (*T_x_*), and liquidus temperature (*T_l_*, obtained from the DSC trace in Figure 1c) of the Tb_55_Co_30_Fe_15_ ribbon are about 561 K, 606 K and 1023 K, respectively. The reduced glass transition temperature (*T_rg_* = *T_g_*/*T_l_*) [12] of the Tb_55_Co_30_Fe_15_ ribbon is therefore found to be about 0.548, and the parameter γ (=*T_x_*/(*T_g_* + *T_l_*)) [13] for the ribbon is 0.383. Compared to the Tb_55_Co_45_ amorphous ribbon, the *T_rg_* of the Tb_55_Co_30_Fe_15_ ribbon is over 20% higher than that of the Tb_55_Co_45_ amorphous ribbon, and the γ value is about 11% higher. As the two parameters are the most commonly used glass formability gauge, the higher *T_rg_* and γ values of the Tb_55_Co_30_Fe_15_ alloy indicate enhanced glass formability by 15% (at. %) Fe substitution for Co.

The relationship between magnetization and temperature (*M*-*T* curves) of the Tb_55_Co_30_Fe_15_ ribbon is revealed in Figure 2a. The sample was firstly cooled from room temperature to 10 K without a magnetic field, and subsequently heated to room temperature under a field of 0.03 T for the measurement of the zero-field-cooled (ZFC) *M*-*T* curve. Then, the sample was cooled from room temperature to 10 K under a magnetic field of 0.03 T and heated to room temperature under the same magnetic field for the measurement of field-cooled (FC) *M*-*T* curve. The Curie temperature (*T_c_*) of the sample obtained from the derivative of the *M*-*T* curves is about 169 K, which is 64 K higher than that of the Tb_55_Co_45_ amorphous ribbon, demonstrating the enhancement of magnetic ordering temperature by 15% (at. %) Fe substitution for Co. The shape of the FC and ZFC *M*-*T* curves of the Tb_55_Co_30_Fe_15_ ribbon is similar to those of canonical spin glass systems [11,14,15,16,17,18,19]. The spin freezing temperature (*T_f_*) obtained from the ZFC *M*-*T* curve is about 139 K, which is also 44 K higher than that of the Tb_55_Co_45_ amorphous ribbon.

It is worth noting that the *M*-*T* curves of the Tb_55_Co_30_Fe_15_ ribbon are much smoother than other metallic glasses, and the magnetization of the sample does not drop to zero even at a temperature far above the Curie temperature. On the other hand, in contrast to other fully amorphous alloys with spin-glass-like behavior [11,14,15,16,17,18,19], the divergence between the ZFC and FC *M*-*T* curves is observed at temperatures well above the Curie temperature. Both the above phenomena indicate that the Tb_55_Co_30_Fe_15_ ribbon may be inhomogeneous in the nanoscale.

Figure 2b shows the hysteresis loops of the Tb_55_Co_30_Fe_15_ ribbon measured at 10 K, 30 K, 50 K, 90 K, 140 K, 170 K, and 300 K. Just like other spin-glass-like metallic glasses [10,14,16,18], the ribbon is hard magnetic at low temperature with a coercivity of ~1.42 T at 10 K and is nearly paramagnetic at 300 K. However, although the coercivity of the sample decreases with the increasing temperature, the sample is not soft magnetic within the temperature range from *T_f_* to *T_c_*. The nearly 0.04 T large coercivity of the sample at 170 K indicates that the Tb_55_Co_30_Fe_15_ ribbon is most likely inhomogeneous in the microstructure because a fully amorphous spin-glass-like alloy should be soft magnetic at temperatures between its *T_f_* and *T_c_* [11,15,19].

In order to ascertain the above assumption, we observed the microstructure of the Tb_55_Co_30_Fe_15_ as-spun ribbon. The HRTEM image of the as-spun ribbon is shown in Figure 3. One can find that there are some nanoparticles with an average diameter less than 10 nm distributed randomly in the disordered matrix. Such microstructure is similar to the structure of Nd-Fe-Al bulk metallic glasses [20,21], which is supposed to be closely related to the positive mixing enthalpy between the Nd and Fe atoms. Using the Miedema’s model [22], we found that the mixing enthalpy between the Tb and Fe atoms is also positive. Therefore, the unique microstructure of the Tb_55_Co_30_Fe_15_ as-spun ribbon with some metastable intermediate nanoparticles embedded in disordered matrix is most likely resulted from the positive mixing enthalpy between the Tb and Fe atoms.

Figure 4a shows the magnetization (*M*-*H*) curves of the Tb_55_Co_30_Fe_15_ as-spun ribbon measured at various temperatures under 5 T. The dramatic decreasing of magnetization under a low magnetic field (<0.5 T) from 90 K to 10 K also illustrates the spin-glass-like behavior of the Tb_55_Co_30_Fe_15_ as-spun ribbon. Based on the isothermal *M*-*H* curves, we can derive the magnetic entropy change (−Δ*S_m_*) at different temperature ((−Δ*S_m_*)-*T* plots) of the Tb_55_Co_30_Fe_15_ ribbon under various magnetic fields, as shown in Figure 4b. Like the situation in other spin-glass-like RE-TM metallic glasses [14,18,19], the high coercivity and the spin freezing behavior obviously undermine the magnetocaloric properties of the Tb_55_Co_30_Fe_15_ ribbon at temperatures below *T_f_*, and the −Δ*S_m_* value is even decreased to negative at temperatures below 50 K. On the other hand, compared to the Tb_55_Co_45_ amorphous ribbon, as shown in Figure 4c, the Tb_55_Co_30_Fe_15_ ribbon shows a much broader and lower −Δ*S_m_* peak. Furthermore, one can find that unlike other fully amorphous alloys, the maximum −Δ*S_m_* (−Δ*S_m_^peak^*) of the Tb_55_Co_30_Fe_15_ ribbon does not appear near its Curie temperature, but appears near 145 K, which is over 20 K lower than the Curie temperature of the ribbon. The deteriorated magnetocaloric properties, as well as the deviation of the −Δ*S_m_^peak^* temperature from *T_c_*, are considered to be induced by the existence of nanoparticles in the amorphous matrix and the resulted residual coercivity near *T_c_* in the Tb_55_Co_30_Fe_15_ ribbon.

The deterioration of the magnetocaloric properties of the Tb_55_Co_30_Fe_15_ ribbon does not necessarily lead to its poor magnetostriction because the magnetocaloric property of RE-TM metallic glasses depends mainly on the interaction between RE and TM atoms, while the magnetoelastic properties of these alloys are dominated by the random magnetic anisotropy, which is closely related to the spin freezing behavior and hysteresis in these alloys. In addition, it is reported that the partial crystallization may be helpful for the improvement of magnetostriction of the Tb(Dy)-Fe amorphous alloy [6,23,24]. Therefore, the microstructure and the high coercivity of the Tb_55_Co_30_Fe_15_ ribbon may lead to a higher magnetostriction. Figure 5 shows the reversible magnetostriction (*λ*-*H*) curves of the Tb_55_Co_30_Fe_15_ ribbon and the Tb_55_Co_45_ amorphous ribbon for comparison, both of which are measured at 50 K. As expected, high values of 520 ppm and 788 ppm are observed under 2 T and 5 T, respectively. These values are much higher than the Tb_55_Co_45_ amorphous ribbon (270 ppm and 465 ppm under 2T and 5 T, respectively), and other Tb(Dy)-Co amorphous ribbons [10,11]. The *λ*-*H* curves of the Tb_55_Co_30_Fe_15_ ribbon were also measured at 90 K, 140 K, and 170 K. The magnetostriction is ~395 ppm at 90 K, ~197.5 ppm at 140 K and ~96 ppm at 170 K. Considering the coercivity of the Tb_55_Co_30_Fe_15_ ribbon decreases from ~0.3 T at 50 K, to ~0.14 T at 90 K, ~0.065 T at 140 K, and 0.04 T at 170 K, one can find that the magnetostriction decreases monotonically with the coercivity, which ascertains the close relationship between the magnetostriction and hysteresis in the Tb_55_Co_30_Fe_15_ ribbon.

## 4. Conclusions

In summary, the microstructure of a Tb_55_Co_30_Fe_15_ glassy ribbon and its effect on the magnetic and magnetostrictive properties were investigated. The ribbon shows typical amorphous characteristics, and its apparent glass formability is better than the binary Tb_55_Co_45_ amorphous ribbon. The Tb_55_Co_30_Fe_15_ ribbon shows typical spin-glass-like feature in its ZFC and FC *M*-*T* curves, with a *T_c_* of ~169 K and a *T_f_* of ~139 K. However, the magnetization in the *M*-*T* curves that does not go to zero at temperatures far above *T_c_*, and the residual coercivity near *T_c_*, indicate that the Tb_55_Co_30_Fe_15_ glassy ribbon may be inhomogeneous in the nanoscale. HREM observation has revealed that there are some nanoparticles with an average diameter less than 5 nm distributed randomly in the matrix. The formation of the nanoparticles is supposed to be due to the positive mixing enthalpy between the Tb and Fe atoms. It was found that the −Δ*S_m_* peak of the nano-glass ribbon is low and the −Δ*S_m_^peak^* does not appear near its Curie temperature. The deterioration of the magnetocaloric property of the ribbon is considered to be induced by the existence of nanoparticles in the amorphous matrix and the resulted residual coercivity near *T_c_*. In contrast to the poor magnetocaloric properties, the nano-glass structure and the resulting high coercivity of the Tb_55_Co_30_Fe_15_ ribbon led to a rather high value of reversible magnetostriction up to 788 ppm under 5 T, which is almost the highest in the amorphous alloys reported in the literature [6,10,11,23,24].

## Figures and Tables

**Figure 1 materials-14-03068-f001:**
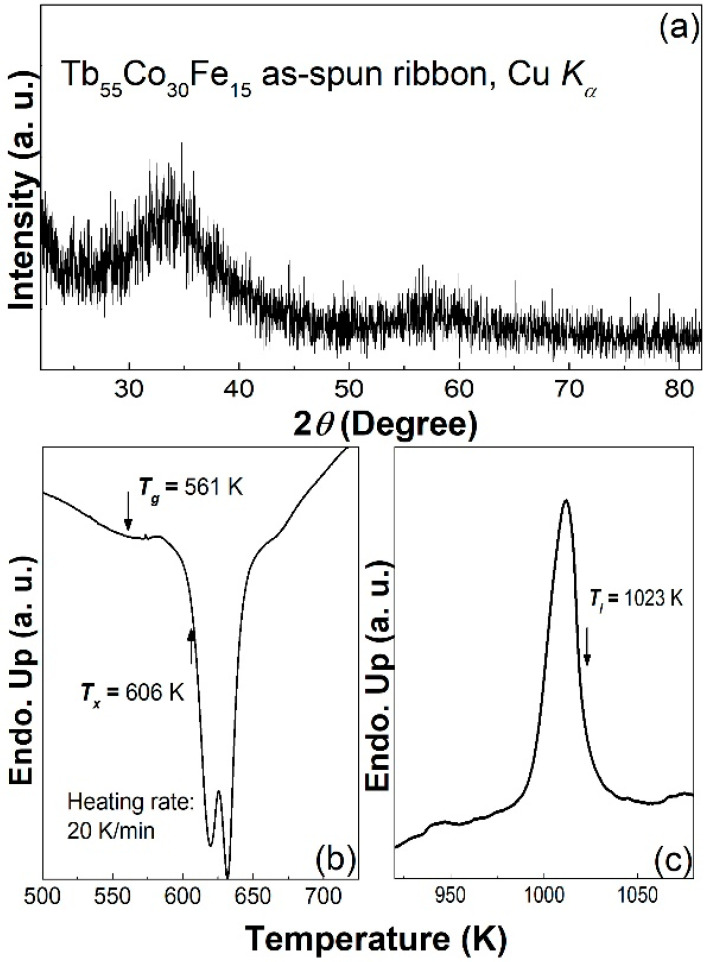
(**a**) XRD pattern of the Tb_55_Co_30_Fe_15_ as-spun ribbon, (**b**,**c**) are the glass transition, crystallization, and melting behaviors of the ribbon.

**Figure 2 materials-14-03068-f002:**
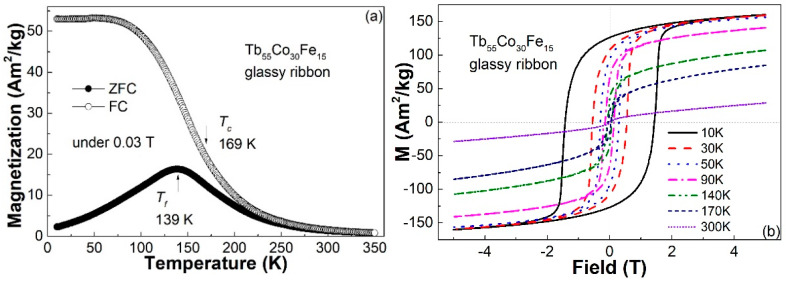
(**a**) ZFC and FC *M*-*T* curves of the Tb_55_Co_30_Fe_15_ glassy ribbon under a magnetic field of 0.03 T; (**b**) the hysteresis loops of the ribbon measured at 10 K, 30 K, 50 K, 90 K, 140 K, 170 K, and 300 K under a field of 5 T.

**Figure 3 materials-14-03068-f003:**
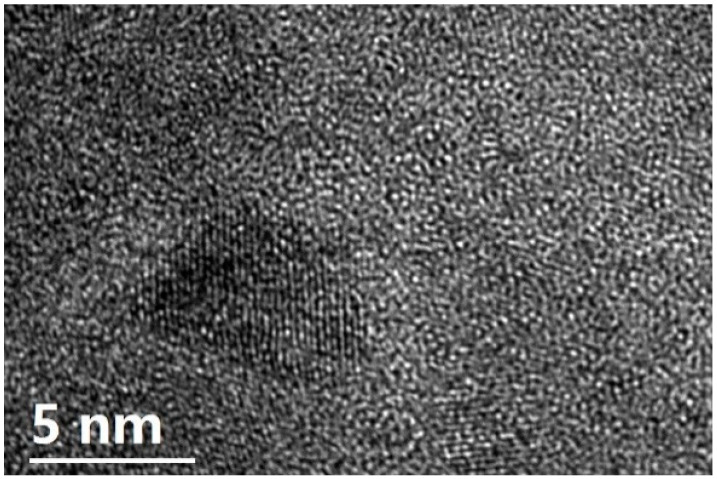
The HRTEM image of the Tb_55_Co_30_Fe_15_ as-spun ribbon.

**Figure 4 materials-14-03068-f004:**
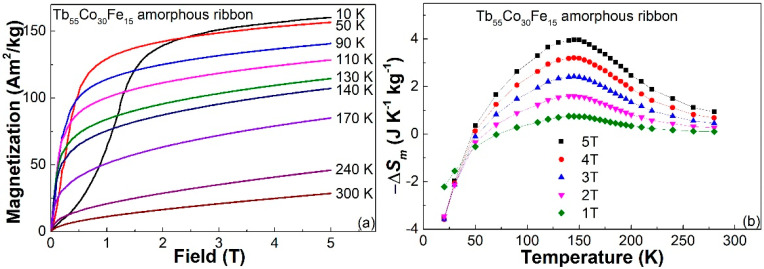
(**a**) The *M*-*H* curves of the Tb_55_Co_30_Fe_15_ amorphous ribbon measured at various temperature range from 10 K to 300 K under 5 T; (**b**) (−Δ*S_m_*)-*T* plots of the Tb_55_Co_30_Fe_15_ ribbon under various magnetic fields; and (**c**) (−Δ*S_m_*)-*T* plots of the Tb_55_Co_45_ and Tb_55_Co_30_Fe_15_ amorphous alloys under 5 T.

**Figure 5 materials-14-03068-f005:**
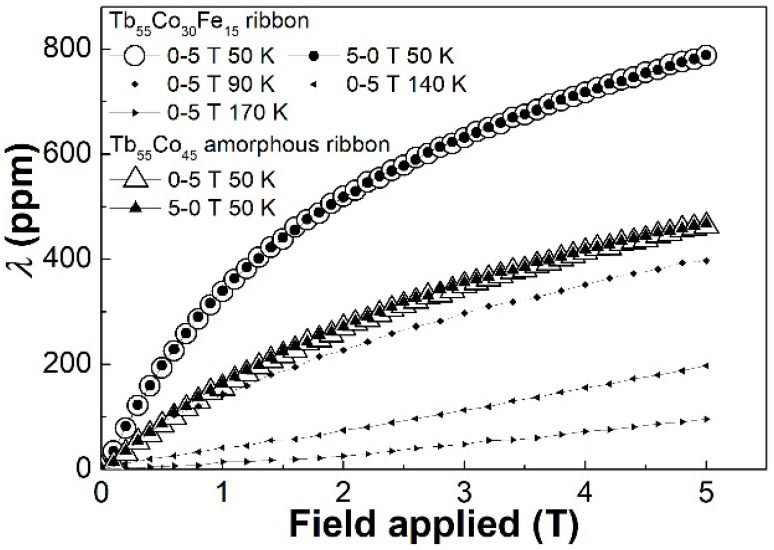
The reversible *λ*-*H* curves of the Tb_55_Co_30_Fe_15_ and Tb_55_Co_45_ amorphous ribbons measured at 50 K under a magnetic field of 5 T, and the *λ*-*H* curves of the Tb_55_Co_30_Fe_15_ amorphous ribbon measured at 90 K, 140 K, and 170 K under a magnetic field of 5 T.

## Data Availability

Data available on the request to the correspondence author.

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
