# Peer review of "Microstructure and Its Effect on the Magnetic, Magnetocaloric and Magnetostrictive Properties of Tb55Co30Fe15 Glassy Ribbons"

_materials, 2021, doi:10.3390/ma14113068_

Round 1

Reviewer 1 Report

The manuscript resulted clearly written and well-organised. I have only some suggestions for the authors:

  1. In figure 1. insets result difficult to appreciate: please make it more readble.
  2. the authors claimed that the Tb55Co30Fe15 is inhomogeneous in its microstructure due to nanoparticles' formation. Could the authors propose a way to overcome this issue?
  3. I suggest the authors to update the references: only one cited article reported a publiation date beyond 2017.  

Reviewer 2 Report

The article deals with a very current and interesting subject of metallic glasses and alloys with magnetostriction. The authors characterize the material they have created, but the work is lacking comparison to other materials and it alone presents a relatively low scientific value. It is not a typical scientific experiment, but an standard characterization of the alloy. Analyzes are valuable for the field of knowledge, but the decision about the publication is left to the editor. Due to the fact that the work is quite simple and basic, I would like to mention a few comments about it:

line - comment
31 - magnetostriction unit is not described consistently (lines 31, 38, 44, etc.)
39 - "very poor" - please provide a quantifiable measure of amorphization (for example - what elements sizes can be made and with what parameters? what is the comparison to other, silimiar alloys?)
42 - Tb-TM and Dy-TM abbreviations introduced without explanation
54 - the introduction is quite brief, I suggest expanding the description of previous works a bit to better introduce the reader to the context of the article
82 - abbreviation Tg without explanation, is this the same as Tg-onset?
88 - graphics absolutely illegible, contains three graphics of different scale without sufficient description and Y axis
124 - the authors analyze the structure of metallic glass and darker areas in the HRTEM image. How to justify that the areas interpreted by the authors are "some nano-particles" - is phase, scattering and diffractive contrast responsible for the darker color? Or could it be connected to sample preparation or over-heating the alloy during polishing?
189 - "(...) the nano-glass structure and the resulted high coercivity of the Tb55Co30Fe15 ribbon lead to the outstanding magnetostrictive properties. The observed high values ​​of the reversible magnetostriction of the Tb55Co30Fe15 ribbon are almost the highest in the amorphous alloys reported in the literatures "- what does "outstanding properties" mean? Comparing magnetostriction to literature values ​​requires reference to the sources.

Reviewer 3 Report

The authors performed an experimental investigation of the magnetic, magnetocaloric, and magnetostrictive properties of Tb55Co30Fe15 glassy ribbons. The possible effect of the existence of microstructure on a nano-scale was discussed. The results are original, of general interests, and have significant implication on how to improve the magnetostriction in this kind of materials. However, there are some significant problems that have to be solved before my recommendation for publication.

  1. The authors claim that the residual coercivity induced by the nano-particle structures is responsible for the magnetostrictive properties. While this is possible, the evidence shown is not sufficient for making this claim. The authors may need to show more evidence about the relationship between coercivity and magnetostriction. For instance, the authors have measured a series of M-H curves in Fig.2  from which a temperature-dependent coercivity can be obtained. If the authors can show more data of magnetostriction at different temperatures, they may establish a correlation between coercivity and magnetostriction. I would be convinced if the authors can make this comparison.
  2. what is the nature of those nano-particles? Is their chemical composition or local crystal structure different from the surrounding background? The divergence between FC and ZFC curves suggest there exists short range magnetic order which is considered as  inhomogeneity. Is the magnetic inhomogeneity related to these structural nano-particles? Can the authors make any comments about these?
  3. In figure,4, the inset in (b) is too small to be read. The authors may take it out as a independent figure to make it clear.

Round 2

Reviewer 2 Report

  • Fig. 1 still not consistent, please rescale/redraw the axis and description
  • Authors see nanoparticles embedded in amorphous matrix but this observation is not confirmed with any other method, but only the citation to "other works" from the same authors. From my experience in metallic glasses sample preparation I sometimes get the same effect after slight over-treatment and over-heating the sample with ion polishing. The authors said that the ion polishing was the only process of sample preparation - due to the slow process of argon ion polishing it probably took a lot of hours, which could be enough to precipitate some parts of the sample. Much more reliable sample preparation is electrolytic polishing, which have no temperature influence, with maybe only additional single-minutes ion cleaning. There is a possibility that authors describe not microstructure itself, but the effect of sample preparation. I am aware that preparation samples from metallic glasses are really tricky, but we should avoid unconfirmed theories.

Reviewer 3 Report

The authors have dealt with all my comments.

I'm satisfied with their responses and recommend this manuscript for publication.

Author Response

Thanks very much for the reviewer's comments.